# Rad27 and Exo1 function in different excision pathways for mismatch repair in *Saccharomyces cerevisiae*

Felipe A. Calil [1], Bin-Zhong Li[1], Kendall A. Torres [1], Katarina Nguyen[1], Nikki Bowen[1], Christopher D. Putnam[1,2] & Richard D. Kolodner [1,3,4,5✉]

Eukaryotic DNA Mismatch Repair (MMR) involves redundant exonuclease 1 (Exo1)-dependent and Exo1-independent pathways, of which the Exo1-independent pathway(s) is not well understood. The *exo1Δ440-702* mutation, which deletes the MutS Homolog 2 (Msh2) and MutL Homolog 1 (Mlh1) interacting peptides (SHIP and MIP boxes, respectively), eliminates the Exo1 MMR functions but is not lethal in combination with *rad27Δ* mutations. Analyzing the effect of different combinations of the *exo1Δ440-702* mutation, a *rad27Δ* mutation and the *pms1-A99V* mutation, which inactivates an Exo1-independent MMR pathway, demonstrated that each of these mutations inactivates a different MMR pathway. Furthermore, it was possible to reconstitute a Rad27- and Msh2-Msh6-dependent MMR reaction in vitro using a mispaired DNA substrate and other MMR proteins. Our results demonstrate Rad27 defines an Exo1-independent eukaryotic MMR pathway that is redundant with at least two other MMR pathways.

[1] Ludwig Institute for Cancer Research, University of California School of Medicine, San Diego, 9500 Gilman Drive, La Jolla, CA 92093-0660, USA. [2] Departments of Medicine, University of California School of Medicine, San Diego, 9500 Gilman Drive, La Jolla, CA 92093-0660, USA. [3] Cellular and Molecular Medicine, University of California School of Medicine, San Diego, 9500 Gilman Drive, La Jolla, CA 92093-0660, USA. [4] Moores-UCSD Cancer Center, University of California School of Medicine, San Diego, 9500 Gilman Drive, La Jolla, CA 92093-0660, USA. [5] Institute of Genomic Medicine, University of California School of Medicine, San Diego, 9500 Gilman Drive, La Jolla, CA 92093-0660, USA. ✉email: rkolodner@health.ucsd.edu

Eukaryotic MMR plays a critical role in suppressing spontaneous mutations caused by DNA replication errors and consequently plays an important role in suppressing the development of cancer[1–8]. The core eukaryotic MMR machinery required for all MMR pathways consists of a mispair recognition complex and an endonuclease active site-containing accessory factor. The mispair recognition complex can be either Msh2-Msh6 or Msh2-Msh3, which have partially overlapping mispair recognition specificities[2,3,7,9–11]. The primary accessory factor in MMR is the Mlh1-Pms1 endonuclease (called MLH1-PMS2 in humans), although a second endonuclease, Mlh1-Mlh3, can substitute for Mlh1-Pms1 to a small extent[2,3,7,12–16]. These proteins promote MMR in combination with appropriate combinations of Exo1, DNA polymerase δ, DNA polymerase ε, PCNA, RFC, and RPA[15,17–22]. The best understood MMR reaction involves recruitment of Exo1 by mispair-bound Msh2-Msh6 (or Msh2-Msh3) to degrade one DNA strand from a preexisting 5' nick or a 5' nick introduced by Mlh1-Pms1 to excise the mispair followed by a gap-filling reaction[15,17–22]. Deletion of *EXO1* only causes a small increase in mutation rate and hence only causes a small MMR defect compared to the deletion of genes that are absolutely required for MMR (e.g., *MSH2*, *MLH1*), implying that Exo1-independent MMR pathways exist[23–25]. Genetic studies have defined MMR excision-resynthesis pathways that depend on or are independent of Exo1[1,23,26,27]. These pathways appear to differ in their requirement for the level of activation of the Mlh1-Pms1 endonuclease, as well as in their requirement for Exo1 recruitment[1,21,26,28]. However, comparatively little is known about the mechanism of Exo1-independent MMR due to the limited state of genetic analysis of potential Exo1-independent MMR mechanisms[1,23,26,29]. Furthermore, since no combination of *exo1Δ* and known mutations that inactivate Exo1-independent MMR pathways completely eliminates MMR[23,26,28,30], it is not clear how many Exo1-independent MMR mechanisms exist.

One candidate for an enzyme that, like Exo1, catalyzes excision during MMR is the flap endonuclease Rad27 (called Rth1 in some studies and FEN1 in humans). Rad27 processes the 5' ends of Okazaki fragments and also acts in base-excision repair[31,32]. A role for Rad27 in MMR was initially suggested because *rad27Δ* strains have a strong mutator phenotype with increased rates of dinucleotide repeat instability and accumulation of mutations in the *CAN1* gene[33]. However, several lines of evidence initially argued against a primary role for Rad27 in MMR. First, defects in *RAD27* primarily increased the rate of insertions in dinucleotide repeats, whereas defects in *MSH2*, *MLH1,* or *PMS1* primarily increased the rate of deletions[33]. Second, combining a deletion of *RAD27* with deletions in *MSH2*, *MLH1* or *PMS1* often resulted in a greater-than-additive increase in the rate of accumulating inactivating mutations in the *CAN1* gene compared to the respective single mutants[33,34], which could reflect defects in more than one DNA repair pathway. Third, defects in *RAD27* caused the accumulation of distinctive large insertion/deletion mutations in *CAN1* that were flanked by short direct repeats that were not caused by a deletion of *MSH2* that causes a complete defect in MMR[35]; increased rates of similar complex mutations, as well as increased rates of genome rearrangements including translocations, were also seen in subsequent studies of *rad27* mutants[34,36,37]. The initial study documenting the occurrence of complex mutations in *rad27* mutants also observed that deletion of *RAD27* caused a small increase in mutation rate in the *hom3-10* frameshift reversion assay[35], which detects frameshift mutations in a mononucleotide repeat sequence and is highly specific for detecting MMR defects[9]; note that MMR defects caused by deletion of genes like *MSH2* cause very large increases in mononucleotide repeat frameshift reversion assays like the *hom3-10* assay[9,35,38–40]. This latter result was interpreted as suggesting

Rad27 could play a minor role in MMR[35]. Biochemical studies have shown that Rad27 can cleave 5' flaps produced by strand-displacement synthesis by DNA polymerase δ on nicked, mispair-containing DNA substrates resulting in mispair excision[29] and Rad27 along with MMR proteins can excise mispairs at the 5' end of Okazaki fragments by DNA polymerase α segment error editing[41]; however, it has not demonstrated that these reactions are related to *bona fide* general MMR. In addition, MMR can act as a complementary pathway to Rad27 in the removal of short 5' flaps that can occur during the processing of the 5' ends of Okazaki fragments[34].

Exo1 and Rad27 catalyze similar reactions. On a 5' nicked substrate, 5' to 3' excision by Exo1 promotes gap filling by DNA polymerase δ, leading to repair of a mispair 3' to the nick[17–20,22]. Similarly, Rad27 will cleave 5' flaps generated during strand-displacement synthesis by DNA polymerase δ on a nicked substrate, resulting in nick translation leading to the same type of reaction products[42,43]. This biochemical similarity suggested it would be useful to investigate the effect of *exo1Δ rad27Δ* double mutations on MMR; however, this combination of mutations is lethal[25,44]. Recently, we observed that the role of Exo1 in MMR is mediated by two Msh2-interacting sequences[28] (SHIP boxes) and one Mlh1-interacting sequence[28,45–47] (MIP box) in the unstructured C-terminus of Exo1[28]. Deleting these sequences results in a mutant Exo1 that is fully defective for MMR but retains its other functions, including those required in the absence of Rad27[28]. Using an *exo1* truncation mutation that eliminates the SHIP and MIP boxes, we have now been able to perform genetic studies that demonstrate that Exo1 and Rad27 play redundant roles in MMR. In addition, we have reconstituted a Rad27-dependent and Msh2-Msh6-dependent MMR reaction in vitro, providing biochemical insight into the mechanism of Rad27-dependent MMR.

## Results

**Genetic evidence that Rad27 and Exo1 function in redundant MMR pathways.** To study genetic interactions between a *rad27Δ* mutation and an MMR-defective *exo1* mutation, we created the *exo1Δ440-702* mutation, which truncates Exo1 after codon 439 and removes the MIP and SHIP boxes that mediate interactions with Mlh1 and Msh2[28]. The *exo1Δ440-702* and *rad27Δ* single and double mutations were tested for their effects on MMR using the *hom3-10* and *lys2-10A* frameshift reversion mutation rate assays, which are highly specific for detecting MMR defects as MMR defects cause high rates of reversion of a single nucleotide insertion in a mononucleotide repeat[9,35,38–40] (Table 1). Consistent with previous studies, the *exo1Δ440-702* mutation caused a small 3-fold increase in mutation rate in both assays, and the *rad27Δ* mutation caused 97-fold and 308-fold increases in mutation rates in the *hom3-10* and *lys2-10A* frameshift reversion assays, respectively. The *exo1Δ440-702 rad27Δ* double mutant had a synergistic increase in mutation rate in both assays compared to both single mutants that were highly significant (Mann−Whitney U-test; Supplementary Table 1). Importantly, the resulting double mutant mutation rate was not as high as that of an *msh2Δ* single mutant, which is completely defective for MMR[9,38,40,48].

To determine the spectrum of mutations that reverted the *hom3-10* mutation, 24 to 36 independent *hom3-10* revertants were sequenced from the *exo1Δ440-702* and *rad27Δ* single and double mutants, as well as from the wild-type and *msh2Δ* strains. The *hom3-10* mutation is caused by a single T insertion in a run of six Ts in the *HOM3* gene[9], and >89% of the reverting mutations from each strain were caused by deletion of a single T in this seven-T run (Supplementary Fig. 1). The remaining

**Table 1 MMR defects caused by genetic interactions between *exo1Δ440-702*, *rad27Δ*, and *pms1-A99V* mutations.**

| Relevant genotype | *hom3-10* reversion rate (×10⁻⁹) | *lys2-10A* reversion rate (×10⁻⁹) |
|---|---|---|
| Wild-type | 1.77 [0.76−4.32] (1) | 5.94 [1.31−13.6] (1) |
| *msh2Δ* | 4200 [3390−7870] (2373) | 136000 [124000−465000] (22896) |
| *pms1Δ* | 4700 [3140−7230] (2655) | 94300 [73300−117000] (15875) |
| *exo1Δ440-702* | 5.33 [4.34−6.54] (3.0) | 17.9 [11.8−23] (3.0) |
| *rad27Δ* | 171 [93−337] (96.8) | 1830 [1310−2920] (308) |
| *exo1Δ440-702 rad27Δ* | 562 [337−753] (317) | 6980 [4130−11600] (1176) |
| *exo1Δ440-702 rad27Δ msh2Δ* | 17700 [8220−45600] (10025) | 244000 [159000−491000] (33216) |
| *exo1Δ440-702 rad27Δ pms1Δ* | 19500 [8910−35800] (11003) | 197000 [103000−573000] (41127) |
| *pms1-A99V* | 4.40 [3.06−5.79] (2.5) | 71.9 [64.8−160] (12.1) |
| *pms1-A99V exo1Δ440-702* | 336 [239−584] (190) | 15200 [11900−18300] (2559) |
| *pms1-A99V rad27Δ* | 1360 [1150−1740] (768) | 22500 [16600−31300] (3316) |
| *pms1-A99V exo1Δ440-702 rad27Δ* | 2260 [1840−2820] (1276) | 43400 [28000−67400] (7307) |

The numbers in square brackets are the 95% confidence intervals. The numbers in parentheses are the fold increase in mutation rate relative to the wild-type strain. Statistical comparison rates are reported in Supplementary Table 1 and Supplementary Table 2. The complete genotypes of the strains used, and the strain identifier numbers are listed in Supplementary Table 3.

reverting mutations were single nucleotide deletions within 10 nucleotides of the seven-T run, often in shorter mononucleotide repeats, which also restored the *HOM3* reading frame (Supplementary Fig. 1). The accumulation of single base deletions and not direct-repeat mediated insertions and deletions is consistent with mutations arising due to MMR defects[35]. Note that previous studies have also shown that the *exo1Δ msh2Δ* and *rad27Δ msh2Δ* double mutation combinations caused the same mutation rate in the *hom3-10* reversion assay as an *msh2Δ* single mutation and, where tested, resulted in the same spectrum of *hom3-10* reverting mutations[24,25,35]. In aggregate, these results and previous results support the hypothesis that Rad27 and Exo1 function in redundant MMR pathways that together only partially account for MMR, which is absolutely dependent on Msh2.

**Genetic evidence for at least two Exo1-independent MMR pathways.** Previous genetic studies have identified mutations inactivating at least one Exo1-independent MMR pathway[1,23,26,27]. The best-characterized mutations affect PCNA (e.g., *pol30-K217E*) and Mlh1-Pms1 (e.g., *pms1-A99V*) and reduce, but do not eliminate, PCNA-activated Mlh1-Pms1 endonuclease activity. This result suggests that at least one Exo1-independent MMR pathway requires higher levels of Mlh1-Pms1 endonuclease activity than Exo1-dependent MMR[1,26,27]. Combining an *exo1Δ* mutation with the *pms1-A99V* or *pol30-K217E* mutations resulted in highly synergistic mutation rates; however, these mutation combinations did not inactivate MMR to the extent caused by an *msh2Δ* mutation[23,26,27]. To determine if Rad27 accounts for the residual Exo1-independent MMR, we combined the *exo1Δ440-702* and *rad27Δ* single and double mutations with the *pms1-A99V* mutation and tested their effects in the *hom3-10* and *lys2-10A* frameshift reversion assays (Table 1). Both the *exo1Δ440-702* and *rad27Δ* single mutations had significant synergistic interactions with the *pms1-A99V* mutation in both assays (Mann Whitney U-test, Supplementary Table 1). In addition, the *exo1Δ440-702 rad27Δ pms1-A99V* triple mutant had a synergistic increase in the mutation rate in the two frameshift mutation rate assays compared to the respective double mutants that were statistically significant (Mann Whitney U-test, Supplementary Table 2). The mutation rates of the *exo1Δ440-702 rad27Δ pms1-A99V* triple mutant were modestly lower than that of an *msh2Δ* single mutant, based on 95% confidence intervals. We then determined the mutation rates of the *msh2Δ exo1Δ440-702 rad27Δ* and *pms1Δ exo1Δ440-702 rad27Δ* triple mutants and found that the mutation rates were slightly higher than the mutation rates of the *msh2Δ* and *pms1Δ* single mutants by 1.5 to 4 fold, although based on 95% confidence intervals, the differences seen in the *lys2-10A* frameshift reversion assay were not significant (Table 1). These results

support the hypothesis that the *exo1Δ440-702*, *rad27Δ*, and *pms1-A99V* mutations inactivate three different redundant MMR pathways that account for most of Msh2-dependent MMR.

To further characterize the role of Rad27 in MMR, we examined the genetic interaction between *rad27Δ* and *pol30-K217E*, which was identified in a screen for PCNA mutations that synergize with an *exo1Δ* mutation in mutator assays; *pol30-K217E* also causes a partial defect in activation of the Mlh1-Pms1 endonuclease[26]. The *lys2-10A* frameshift reversion rates of the *rad27Δ* and *pol30-K217E* single mutants and the *rad27Δ pol30-K217E* double mutant were $1.83 \times 10^{-6}$ (308 × wild-type), $3.26 \times 10^{-7}$ (55 × wild-type), and $1.02 \times 10^{-5}$ (1717 × wild-type), respectively. Thus, combining the *rad27Δ* and *pol30-K217E* mutations resulted in a significant synergistic increase in mutation rates compared to the respective single mutations ($p = 5.3 \times 10^{-9}$ and $p = 8.6 \times 10^{-10}$, respectively; Mann−Whitney U-test), similar to the genetic interaction between *exo1Δ* and *pol30-K217E*[26], *exo1Δ* and *pms1-A99V*[23], and *rad27Δ* and *pms1-A99V* (Table 1).

**Deletion of *RAD27* results in the accumulation of Mlh1-Pms1 foci MMR intermediates.** Previous studies have shown that Mlh1-Pms1, detected using a functional Pms1-GFP fusion protein, forms foci during S-phase and that the formation of these foci requires the presence of mispaired bases and *MSH2*, and are promoted by the Msh2-Msh6 and Msh2-Msh3 complexes, leading to the conclusion that Mlh1-Pms1 foci are key MMR intermediates[26,27,30,49]. These Mlh1-Pms1 foci accumulate at increased levels in *exo1Δ* mutants, suggesting that Exo1-dependent MMR may be more rapid than Exo1-independent MMR or may require less Mlh1-Pms1[30]. We, therefore, investigated the effect of *exo1Δ440-702* and *rad27Δ* mutations on the levels of Pms1 foci (Fig. 1). The *exo1Δ440-702* and *rad27Δ* single mutants had increased levels of Pms1 foci that were essentially the same as and slightly higher than those observed in an *exo1Δ* mutant. The *exo1Δ440-702 rad27Δ* double mutant had a modest but significantly higher increase in the level of Pms1 foci than the respective single mutants. Importantly, an *msh2Δ* mutation eliminated the Pms1 foci in all of the mutants tested. These results suggest that Rad27-dependent MMR and Exo1-dependent MMR have similar mispair excision kinetics and/or require similar levels of Mlh1-Pms1[30].

**Reconstitution of Rad27-dependent MMR using purified proteins.** Previously we reconstituted an MMR reaction on circular DNA substrates containing different mispairs and a 5' nick[17,50]. These reactions used *S. cerevisiae* Msh2-Msh6, Exo1, DNA

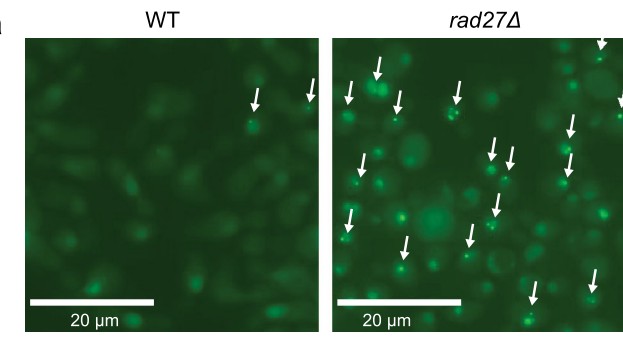

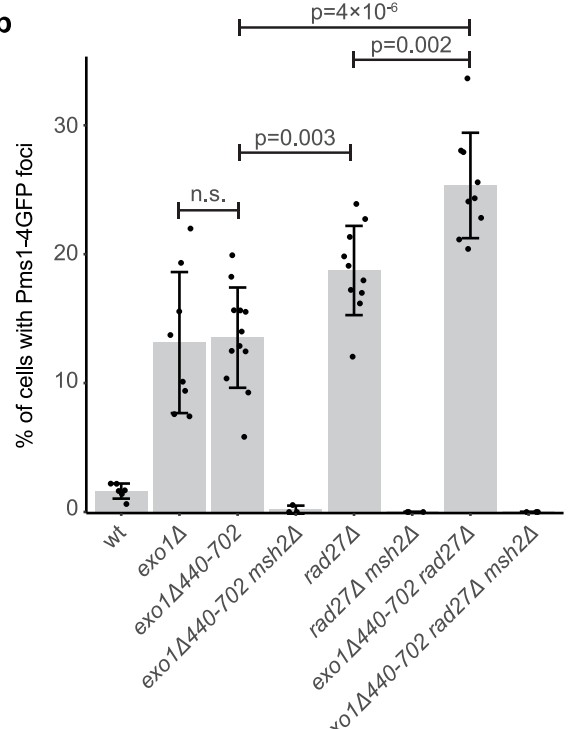

**Fig. 1 Loss of Rad27 causes increased levels of Pms1 foci. a** Example fluorescence micrographs of wild-type and *rad27Δ* cells expressing Pms1-4GFP show increased levels of foci (white arrows) in the *rad27Δ* cells. The bar indicates 20 μm. **b** Quantitation of the average number of Pms1-4GFP foci; error bars represent the standard deviation, the bars indicate the mean of the observations, and the points show the results from each field quantified. The statistical test used was a two-tailed *t*-test. Deletion of *EXO1* or the *exo1Δ440-702* truncation cause equivalent levels of Msh2-dependent Pms1-4GFP foci. Deletion of *RAD27* causes higher levels of Msh2-dependent of Pms1-4GFP foci than the *exo1Δ440-702* truncation, and the *exo1Δ440-702 rad27Δ* double mutant strain has higher levels of Msh2-dependent Pms1-4GFP foci than either single mutant. The reported experiments were done twice and a minimum of 8 independent fields were captured and quantified per strain analyzed.

polymerase δ, PCNA, RPA, and RFC-Δ1N, which is a version of Replication Factor C in which the ligase homology domain of Rfc1 was deleted to allow for overexpression[51]. Repair of the CC mispair-containing substrate tested here allows PstI to cleave at a site that is cleavage-resistant due to the mispair (Fig. 2a). PstI cleavage of the repaired substrate combined with cleavage at a distal ScaI site resulted in characteristic 1.8 and 1.1 kb products (Fig. 2b, lane 1). Omitting Exo1 in these reactions almost completely abolished repair; however, in the small amount of apparent repair products observed, the larger product species

appeared to migrate more slowly than the normal 1.8 kb product (Fig. 2b, lane 2). It seemed likely that this apparent repair was due to strand-displacement synthesis by DNA polymerase δ and that the slowly migrating product contained a branch due to the displaced strand (Fig. 2a); in previous studies, these products were not seen when Exo1 was omitted because we used lower amounts of DNA polymerase δ[17]. To investigate this possibility, different amounts of Rad27 were added to the reactions (Fig. 2b). Increasing amounts of Rad27 markedly stimulated the repair reaction and reduced the aberrant migration of the large product species; the highest levels of Rad27 caused the large product species to co-migrate with the mature 1.8 kb repair product. Stimulation of DNA polymerase δ by Rad27 has been previously observed using different assays[42]. In repair reactions with levels of DNA polymerase ε that promote repair in the presence of Exo1[18], no apparent repair was seen at any amount of Rad27 tested (Fig. 2c); this is consistent with the fact that DNA polymerase ε does not catalyze strand-displacement DNA synthesis[52].

To investigate the requirements for Rad27 and Msh2-Msh6 in these reactions, a series of reactions were performed containing 5 pmoles of Rad27, Msh2-Msh6, DNA polymerase δ, PCNA, RFC-Δ1N, and RPA in which different proteins were omitted as indicated (Fig. 3). In the presence of both Rad27 and Msh2-Msh6, both the mature 1.1 and 1.8 kb species were observed, similar to the Exo1-dependent reaction. In the absence of both Rad27 and Msh2-Msh6, the mature 1.1 kb species and the slowly migrating branched species resulting from strand-displacement synthesis were observed. In the absence of Msh2-Msh6 but in the presence of Rad27, a mixture of the strand-displacement synthesis product and 1.8 kb mature repair species were observed, consistent with the cleavage of a fraction of the strand displacement products by Rad27. Remarkably, in the absence of Rad27 but in the presence of Msh2-Msh6, essentially no products were observed, indicating that Msh2-Msh6 suppresses strand-displacement synthesis by DNA polymerase δ in the absence of Rad27. Thus, the formation of only mature repair products was dependent on both Rad27 and Msh2-Msh6 under these reaction conditions, indicative of a *bona fide* MMR reaction.

**Mispair and PCNA binding by Msh2-Msh6 function in Rad27-dependent MMR.** We then tested the effect of substituting wild-type Msh2-Msh6 with Msh2-Msh6-F337A, which is defective for mispairing binding[53], or Msh2-Msh6-F33AF34A, which is defective for binding PCNA[54] (Fig. 4). Msh2-Msh6-F337A completely eliminated repair, consistent with a requirement for mispair binding by Msh2-Msh6 for the ability of Rad27 to overcome Msh2-Msh6-mediated suppression of strand-displacement synthesis. Msh2-Msh6-F33AF34A resulted in an increased level of apparent repair compared to that seen in the presence of wild-type Msh2-Msh6, and the same distribution of branched and mature repair product bands as that seen when Msh2-Msh6 was omitted from the reactions. This latter result suggests that the interaction between Msh2-Msh6 and PCNA is required to stimulate the formation of mature repair products.

Given the dramatic changes caused by mutant Msh2-Msh6 complexes in the Rad27-dependent MMR reactions, we investigated the effects of these mutant complexes in the absence of Rad27 (Fig. 5). Only strand displacement products were formed in reactions containing DNA polymerase δ but no Rad27 or Msh2-Msh6, as previously observed (Fig. 4). The addition of Msh2-Msh6 suppressed the formation of strand displacement products, as was also previously observed (Fig. 3). The mispair-binding defective Msh2-Msh6-F337A mutant eliminated the formation of strand displacement products, similar to its effect in the presence of Rad27 (Fig. 4), and indicates that mispair

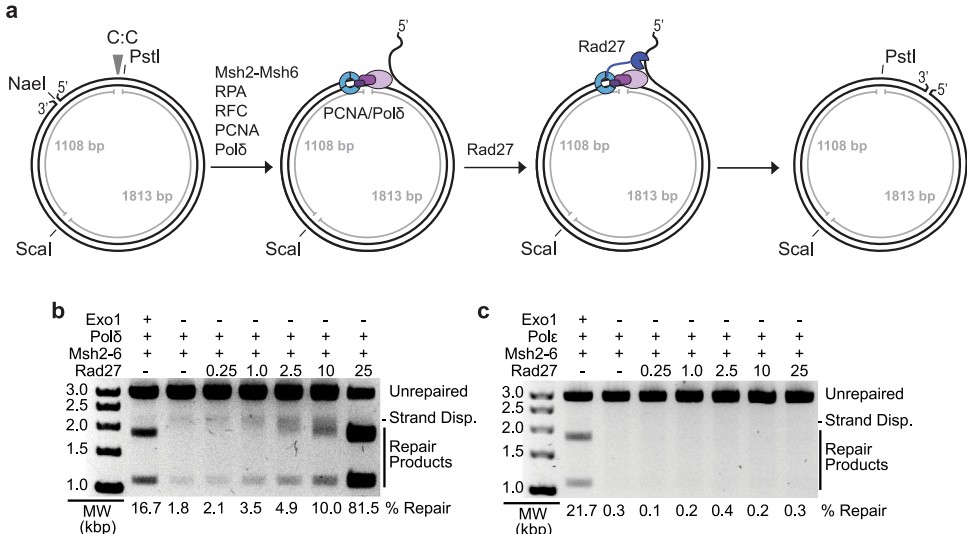

**Fig. 2 Reconstitution of 5′ nick-directed Rad27-dependent MMR using the NaeI-nicked CC substrate. a** Schematic representation of the DNA substrate for the reconstituted repair assays. The substrate has a nick at the NaeI site and the CC mispair, indicated by the arrowhead, disrupts the PstI site. Recruitment of PCNA and DNA polymerase δ to the NaeI nick leads to strand-displacement synthesis and formation of a 5′ flap which is cleaved by Rad27. Incomplete flap cleavage gives rise to slower migrating product bands. **b, c** Assays of 5′ nick-directed repair of the CC substrate, in which different proteins were omitted or substituted as indicated, using either polymerase δ **b** or polymerase ε **c**. Amounts in pmole of Rad27 used are indicated for each lane. All reactions also contained PCNA, RFC-Δ1N, and RPA. Other than Rad27, the amounts of all proteins used in the assays are as listed in the "Methods" section. MW, molecular weight markers. The percent repair corresponds to the fraction of all DNA in each individual lane susceptible to PstI cleavage, including those labeled "strand displacement" and "repair products". A minimum of three independent experiments was performed.

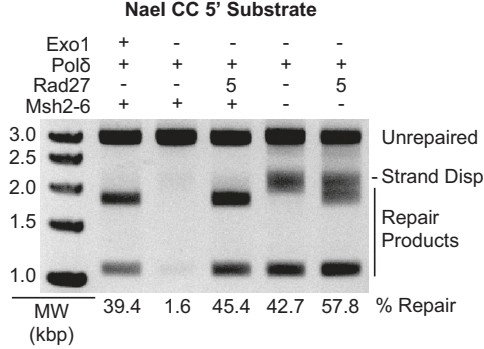

**Fig. 3 Both Rad27 and Msh2-Msh6 are required to form only mature repair product bands.** Assays of 5′ nick-directed repair of the CC substrate, in which different proteins were omitted or substituted as indicated. The amount of Rad27 used is indicated in pmole. All reactions also contain PCNA, RFC-Δ1N, and RPA. Other than Rad27, the amounts of all proteins used in the assays are as listed in the "Methods" section. MW, molecular weight markers. The percent repair corresponds to the fraction of all DNA in each individual lane susceptible to PstI cleavage, including those labeled "strand displacement" and "repair products". Three independent experiments were performed.

binding is not required to suppress strand displacement synthesis. The PCNA-binding defective Msh2-Msh6-F33AF34A mutant resulted in only strand displacement products, although the overall level of these products was lower than seen with DNA polymerase δ alone (Fig. 5). This mirrors the effect of the Msh2-Msh6-F33AF34A mutant in the presence of Rad27 (Fig. 4) and demonstrates that the Msh2-Msh6-PCNA interaction is required for Msh2-Msh6 to suppress the strand displacement reaction catalyzed by DNA polymerase δ.

To gain further insight into the inhibition of DNA synthesis by Msh2-Msh6, experiments were performed in which either the amount of Msh2-Msh6 was decreased or the amount of PCNA

was increased in reactions containing DNA polymerase δ, RFC-Δ1N and RPA (Supplementary Fig. 2). Decreasing the amount of Msh2-Msh6 resulted in the restoration of DNA synthesis-mediated repair products, although ratios of 0.77 and 0.38 molecules of Msh2-Msh6 per PCNA trimer still resulted in some inhibition of DNA synthesis-mediated repair products. Similarly, increasing PCNA to amounts that were greater than the amount of Msh2-Msh6 restored the formation of DNA synthesis-mediated repair products. These results are consistent with the previous observation (see above) that the interaction between the Msh6 subunit of Msh2-Msh6 and PCNA is critical to coordinate Rad27-dependent MMR in vitro and suppress the strand displacement reaction catalyzed by DNA polymerase δ.

**The circular products of Rad27-dependent MMR contain nicks that can be sealed with DNA ligase.** To determine if the circular products of Rad27-dependent MMR were nick-containing plasmids that could be sealed by DNA ligase, we studied the repair of a DNA substrate containing a CC mispair in which the 5′ phosphate present at the NaeI site was removed so that ligatable products could be distinguished from the non-ligatable starting substrate (Supplementary Fig. 3). Dephosphorylation of the substrate was not expected to affect MMR in vitro because it retains the normal 3′ hydroxyl terminus extended by DNA polymerase δ during the repair. In control reactions, treatment of the dephosphorylated substrate with either T4 DNA ligase or T4 DNA ligase followed by treatment with DNA gyrase did not convert any of the DNA to covalently closed circular supercoiled (CCC SC) DNA (Supplementary Fig. 3a). In contrast, treatment of substrate that had not been dephosphorylated with T4 DNA ligase converted a large proportion of the DNA to forms that migrated more rapidly on agarose gels than the nicked circular substrate, and treatment with T4 DNA ligase and DNA gyrase converted most of the substrate to a single, more rapidly migrating CCC SC DNA species (Supplementary Fig. 3b); thus we used treatment with T4 DNA ligase followed by treatment with

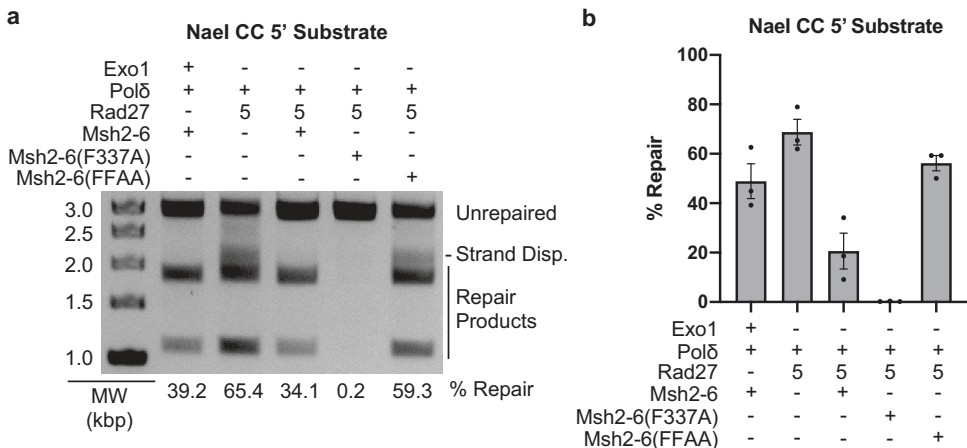

**Fig. 4 Effect of substituting wild-type Msh2-Msh6 for two different mutant Msh2-Msh6 complexes in a 5′ nick-directed Rad27-dependent MMR.**
**a** Representative assay of 5′ nick-directed repair of the CC substrate in which different proteins were omitted or substituted as indicated. All reactions also contain PCNA, RFC-Δ1N, and RPA. Other than Rad27, the amounts of all proteins used in the assays are as listed in the "Methods" section. MW, molecular weight markers. The percent repair corresponds to the fraction of all DNA in each individual lane susceptible to PstI cleavage, including those labeled "strand displacement" and "repair products". **b** The amount of repair for the reactions was quantified. Three independent experiments were performed and the average amount of repair observed is reported. Error bars represent the standard error of the mean, and individual values from the different experiments are indicated by the dots on each histogram bar. Bar graph was constructed in GraphPad Prism software.

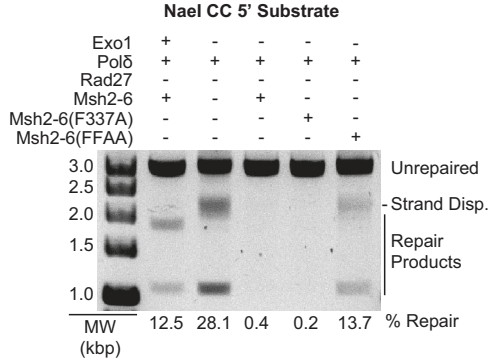

**Fig. 5 Effect of substituting wild-type Msh2-Msh6 for two different mutant Msh2-Msh6 complexes when omitting Rad27.** Representative assay of 5′ nick-directed repair of the CC substrate in which different proteins were omitted or substituted as indicated. All reactions also contain PCNA, RFC-Δ1N, and RPA. The amounts of all proteins used in the assays are as listed in the "Methods" section. MW, molecular weight markers. The percent repair corresponds to the fraction of all DNA in each individual lane susceptible to PstI cleavage, including those labeled "strand displacement" and "repair products". Three independent experiments were performed.

DNA gyrase as an assay for circular DNAs that contained nicks that could be sealed with DNA ligase.

DNA repair reactions were performed under our established conditions except that they contained the CC mispaired substrate with the dephosphorylated 5′ nick at the NaeI site and high amounts of Rad27 minus and plus Msh2-Msh6 (Supplementary Fig. 3c). Under these conditions, a large proportion of the substrate was converted to repair products in reactions lacking Msh2-Msh6, and a somewhat reduced proportion of the substrate was converted to repair products in reactions containing Msh2-Msh6; in both cases, the repair products lacked a 5′ displaced strand. Treatment of the repair products with T4 DNA ligase and DNA gyrase converted a large proportion of the repair products to the CCC SC DNA species indicating that mature repair products were formed that contained nicks that could be sealed with DNA ligase (Supplementary Fig. 3d). The repair was then

examined at a reduced amount of Rad27 resulting in lower repair where in the absence of Msh2-Msh6 the repair products were a mixture of mature and strand displaced products and in the presence of Msh2-Msh6 the repair products were almost exclusively mature repair products (Supplementary Fig. 3e) as previously seen. In both cases, treatment with T4 DNA ligase and DNA gyrase converted a proportion of the products to the CCC SC DNA species (Supplementary Fig. 3f). These results indicate that at a minimum, a large proportion of the products of Rad27-dependent MMR in vitro are circular DNA molecules containing a nick that is a substrate for DNA ligase; this is similar to that seen in Exo1-dependent MMR reactions reconstituted with human MMR proteins[22].

## Discussion
Genetic and biochemical studies have demonstrated the existence of both Exo1-dependent and Exo1-independent MMR pathways[1,23,26,27]. Here we constructed an *exo1* separation-of-function mutation, *exo1Δ440-702*, that inactivates the ability of Exo1 to function in MMR but not the ability of Exo1 to maintain cell viability in the absence of Rad27[28]. This *exo1* mutation allowed us to demonstrate that Rad27 can support MMR in the absence of Exo1-dependent and other known Exo1-independent MMR pathways in vivo[21,28]. Defects in *RAD27* resulted in the accumulation of Mlh1-Pms1 MMR foci intermediates similar to defects in *EXO1*[27,30], suggesting that the Exo1-dependent MMR and Rad27-dependent MMR have similar kinetic properties[30]. Finally, we reconstituted MMR reactions in vitro with a combination of Msh2-Msh6, Rad27, DNA polymerase δ, PCNA, RFC-Δ1N, and RPA. The formation of mature MMR products was dependent on Msh2-Msh6, Rad27, and DNA polymerase δ, as well as the ability of Msh2-Msh6 to both bind mispairs and interact with PCNA through the PCNA interaction site in the N-terminus of Msh6. These studies establish the existence of a general Rad27-dependent MMR pathway.

Most of the mutations that arise in *rad27Δ* mutants are due to mutagenesis mechanisms other than loss of MMR[35–37,55]; however, a minor role for Rad27 in MMR was postulated due to the observation that *rad27Δ* mutations caused modest increases in the rate of mutations detected using MMR-specific frameshift mutation

reversion assays[35]. Here, we found that combining a *rad27Δ* mutation with the *exo1Δ440-702* mutation caused synergistic increases in mutation rates in frameshift reversion assays that are highly specific for MMR defects[9,35,38]. In addition, the *exo1Δ440-702 rad27Δ* double mutant had increased levels of Pms1 foci relative to the respective single mutants. These data are consistent with redundant roles for Rad27 and Exo1 in MMR. The mutation rates caused by the *exo1Δ440-702* and *rad27Δ* single and double mutations were further increased by the addition of the *pms1-A99V* mutation, and the mutation rates caused by the *rad27Δ* mutation was also further increased by the addition of the *pol30-K217E* mutation. Both *pms1-A99V* and *pol30-K217E* were previously shown to enhance the weak MMR defect caused by an *exo1Δ* mutation[23,26]. The mutation rate of the *rad27Δ exo1Δ440-702 pms1-A99V* triple mutant was close to but slightly lower than that of *msh2Δ* and *pms1Δ* single mutants. In addition, the mutation rates of the *msh2Δ exo1Δ440-702 rad27Δ* and *pms1Δ exo1Δ440-702 rad27Δ* triple mutants were similar to the mutation rates of the *msh2Δ* and *pms1Δ* single mutants, respectively. In aggregate, these results suggest the presence of at least three MMR excision pathways (Fig. 6): (1) Exo1-dependent excision, (2) Rad27-dependent excision, and (3) a pathway that is independent of Exo1 and Rad27 but is promoted by elevated levels of Mlh1-Pms1 endonuclease activity, which we have previously called the Exo1-independent pathway[1,21,23,26,27]. Together, these three pathways account for most of Msh2-dependent MMR.

To understand the mechanism of Rad27-dependent MMR, we reconstituted a Rad27-dependent MMR reaction in vitro using Msh2-Msh6, Rad27, DNA polymerase δ, PCNA, RFC-Δ1N and RPA and a mispaired DNA substrate containing a 5' nick at a NaeI site. In the absence of Msh2-Msh6, DNA polymerase δ catalyzes strand displacement synthesis past the mispair, and Rad27 stimulates this synthesis, as well as cleaving off a portion of the branches produced by strand displacement synthesis by DNA polymerase δ. When Msh2-Msh6 was added, the strand displacement activity of DNA polymerase δ was eliminated unless Rad27 was present. In the presence of all three proteins, a coupled strand-displacement and flap-excision reaction were mediated such that only mature MMR products lacking DNA flaps and containing ligatable nicks were formed, consistent with a *bona fide* MMR reaction. The Msh2-Msh6-F337A mutant protein, which cannot bind mispairs[53], retained the ability to eliminate strand-displacement synthesis but failed to coordinate the coupled reaction involving Rad27. In contrast, reactions containing the Msh2-Msh6-F33AF34A mutant protein, which lacks PCNA binding[54], behaved as if Msh2-Msh6 was absent from the reactions, suggesting that the coupled reaction and inhibition of the strand displacement activity of DNA polymerase δ is coordinated by Msh2-Msh6 through PCNA[42,56]; the results of titration experiments in which the amounts of PCNA and Msh2-Msh6 were varied support this model. In aggregate, these results are consistent with a model (Fig. 6) in which mispair-bound Msh2-Msh6 targets DNA polymerase δ-PCNA, Rad27-PCNA, or DNA polymerase δ-Rad27-PCNA complexes to the repair reaction through the Msh6 interaction with PCNA[42,56]. This model is consistent with previous co-precipitation experiments showing that human Msh2-Msh6 interacts directly or indirectly with FEN1[41] (the homolog of Rad27). We have not yet established whether the Rad27 excision pathway can be coupled to mispair-dependent strand nicking by the Mlh1-Pms1 endonuclease, but this seems likely because 5' nicks produced by Mlh1-Pms1 appear to act in a matter that is indistinguishable from pre-existing 5' nicks[15,21,29].

In summary, our results provide support for the hypothesis that Rad27 functions in a nick-directed Msh2-Msh6-dependent mispair excision pathway. In contrast to the Exo1-dependent pathway where a gap is generated and can then be filled by either DNA polymerase δ or DNA polymerase ε, Rad27 appears to act on the 5' flap resulting from strand displacement synthesis by DNA polymerase δ and hence cannot promote repair by DNA polymerase ε, which does not perform strand-displacement synthesis[52]. Our results also support the hypothesis that at least three redundant MMR excision pathways account for all of or the majority of Msh2-dependent MMR (Fig. 6). Recent studies have suggested that strand displacement synthesis by DNA polymerase δ can potentially contribute to Exo1-independent MMR[29] and have shown that FEN1 along with MMR proteins can excise mispairs at the 5' end of Okazaki fragments by DNA polymerase α segment error editing that acts at the 5' ends of Okazaki fragments[41]. Our results extend the results of these prior studies and also demonstrate that Rad27-mediated excision is a general mispair excision pathway like Exo1-mediated excision and is a *bona fide* MMR pathway that functions in vivo.

## Methods

**Strains and plasmids**. *S. cerevisiae* strains were grown in YPD (1% yeast extract, 2% bacto peptone, and 2% dextrose) or in the appropriate synthetic dropout media (0.67% yeast nitrogen base without amino acids, 2% dextrose, and amino acid dropout mix at the concentration recommended by the manufacturer (US Biological) at 30 °C. All transformations with plasmids or PCR-based deletion cassettes were performed using standard lithium acetate transformation protocols.

All *S. cerevisiae* strains used for mutation rate analysis were derived from the S288C strain RDKY5964 **MATa** *ura3-52 leu2Δ1 trp1Δ63 his3Δ200 hom3-10 lys2::InsE-A10* and all strains used for microscopy studies were derived from RDKY7588, which is RDKY5964 containing the *PMS1-4GFP::kanMX6* allele[30]. Mutations were introduced into these strains using standard gene disruption methods (Supplementary Table 3). The PCR primers used in gene disruption experiments are listed in Supplementary Table 4. The *exo1Δ440-702* mutation was introduced into strains using a gene disruption cassette made by amplifying selectable genes from the pFA6a series of plasmids using the PCR primers (uppercase bases correspond to *EXO1* sequence and lowercase sequences correspond to pFA6a plasmid sequence) Primer 1 upstream 5'-CAA TTA AGC GTA GGA AAT TAA GTA ATG CCA ATG TAG TCC AAG AAA CG tag atg aat aac gta cgc tgc agg tcg ac, where the first three bases in lower case are the stop codon that replaces *EXO1* codon 440, and Primer 2 downstream 5'-TTT ACT GGG CAT TGA TTT TTT AAT TCT TGT CTT GAG GCA TTT CGA CGA GAT atc gat gaa ttc gag ctc g. The PCR primers used for generating the *exo1Δ* cassette were Primer 3 upstream 5'-GGT CTA GTA CAA TGG CTT TTT CCC AAA GTA GAA GGC TTC TTA CTC CAA CCG TAC CCT Gcg tac gct gca ggt cga c and Primer 4 downstream 5'-GAC AAT GGC AAT TAA GCG TAG GAA ATT AAG TAA TGC CAA TGT AGT CCA AGA AAC Gta gat gaa taa atc gat gaa ttc gag ctc g. The PCR primers used for generating the *rad27Δ* cassette were Primer 5 upstream 5'-TAT GCC AAG GTG AAG GAC CAA AAG AAG AAA GTG GAA AAA GAA CCC CCt cat cct gat gcg gta ttt tct cc and Primer 6 downstream 5'-CAG CAT ACA TTG GAA AGA AAT AGG AAA CGG ACA CCG GAA GAA AAA ATa tgc gtt tcg gtg atg acg gtg. Disruption mutations were verified by PCR amplification with appropriate upstream, downstream, and internal PCR primers.

**Mutation rate analysis**. Mutator phenotypes were evaluated using the *hom3-10* and *lys2-10A* frameshift reversion assays essentially as previously described[23,30]. Qualitative analysis was done by patching colonies onto YPD plates and replica plating onto -Thr synthetic dropout media for analysis of papillae growth[9,23]. Mutation rates were determined by fluctuation analysis using a minimum of 2 independently derived strains and 14 or more independent cultures per experiment in each of at least 2 independent experiments and calculated using Microsoft Excel software; comparisons of mutation rates were evaluated using 95% confidence intervals or by Mann-Whitney two-tailed tests[23,30].

**Mutation spectra analysis**. One independent Thr+ revertant was isolated per culture from fluctuation tests or per patch from qualitative tests[9]. Chromosomal DNA was isolated from each revertant using a Qiagen Puregene Yeast/Bact. Kit B and the *hom3-10* region were amplified by PCR using the Primer 7 5'-AGTTGTTTGTTGATGACTGC and Primer 8 5'-TTCAGAAGCTTCTTCTGGAG and sequenced with the Primer 9 5'-CTTTCCTGGTTCAAGCATTG (Supplementary Table 4) using a commercial sequencing facility[9].

**Protein purification**. MMR proteins were purified according to standard protocols as previously described for Exo1[17,18], Msh2-Msh6[54,57], PCNA[17,58], DNA polymerase δ[59], RFC-Δ1N[51], and RPA[60] and were greater than 95% pure as determined by SDS-PAGE. Multiple protein preparations were used during the course of the experiments presented and many of these preparations were validated in our previously published studies[17,18,21,26,28].
Rad27 was purified as follows. The *S. cerevisiae RAD27* gene was codon-optimized for expression in *E. coli* (Integrated DNA Technologies) and inserted into the pET28a

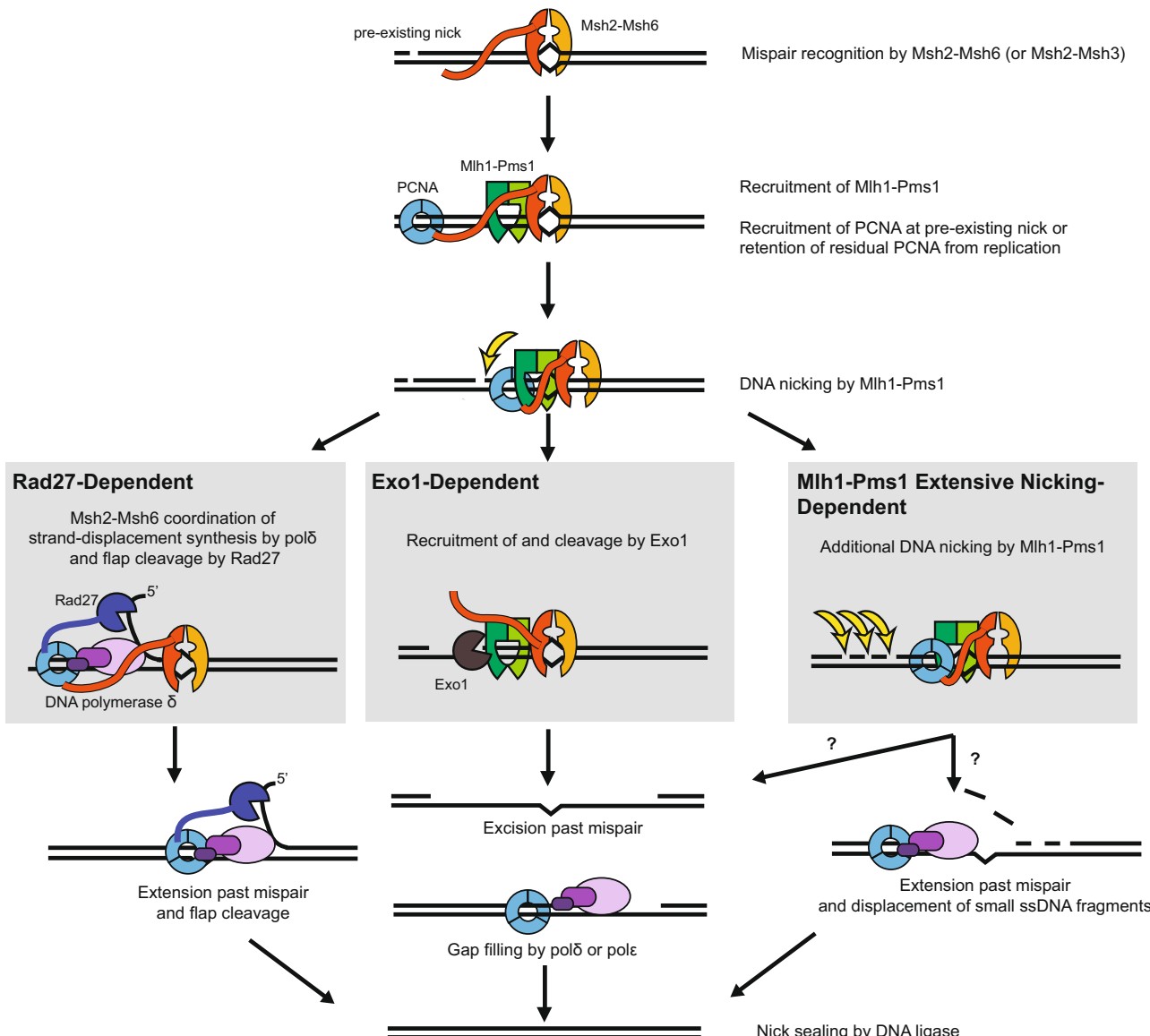

**Fig. 6 Model for different MMR excision-resynthesis pathways.** After mispair recognition by Msh2-Msh6 (or Msh2-Msh3) and recruitment of Mlh1-Pms1, a repair can be initiated by several pathways. In the well-understood Exo1-dependent pathway, recruitment of Exo1 by Msh2-Msh6 and/or Mlh1-Pms1 allows for 5'->3' resection of the nicked strand past the mispair to generate a gap. In the Rad27-dependent pathway, strand-displacement synthesis from a nick by DNA polymerase δ and 5' flap cleavage by Rad27 is coordinated by Msh2-Msh6, likely through shared interactions with PCNA. In both of these pathways, the nicks could be pre-existing or could be made by the Mlh1-Pms1 endonuclease. The less well-understood Exo1-independent and Rad27-independent pathways are hypersensitive to reduced activation of the Mlh1-Pms1 endonuclease, suggesting a role for additional rounds of nicking by Mlh1-Pms1. Additional nicking could generate a gap, equivalent to the Exo1-dependent pathway, or small fragments that can readily be displaced by the strand-displacement synthesis in the absence of Rad27.

vector. *E. coli* OneShot BL21 (DE3) pLysS cells (Invitrogen) were freshly transformed with the resulting expression vector pET28a/Rad27 (pRDK1913) before use. Two liters of culture were grown in LB (1% bacto peptone, 0.5% yeast extract, and 0.5% NaCl), 100 μg/mL kanamycin and 34 μg/mL chloramphenicol at 30 °C until the cell density reached 0.6, and then the expression of Rad27 was induced by addition of isopropyl β-D-thiogalactopyranoside (IPTG) at a final concentration of 0.5 mM for 5 h. The cells were harvested by centrifugation, resuspended in Buffer $A_{100}$ consisting of Buffer A [20 mM Tris, pH 8.0, 0.5 mM ethylenediaminetetraacetic acid (EDTA), 10% (v/v) glycerol, 1 mM dithiothreitol (DTT), protease inhibitor mixture PIC D, (final concentrations of 1 mM phenylmethylsulfonyl fluoride (PMSF), 1 μg/L chymostatin, and 1 μg/L pepstatin A), protease inhibitor mixture PIC W (final concentrations of 1 mM benzamadine, 0.5 μg/L bestatin, 1 μg/L aprotinin, and 1 μg/L leupeptin)] containing 100 mM NaCl and lysed by sonication. The lysate was then clarified by centrifugation at 39,190 × *g* in a Beckman JA-25.50 Rotor at 4 °C for 1 h. The supernatant was then mixed for 1 hr with 72 mL (4 portions in 50 mL centrifuge tubes) of DEAE Sepharose that had been previously equilibrated with Buffer $A_{100}$ and then the resin slurry was centrifuged and the supernatant was saved. The supernatant was

centrifuged a second time to remove any residual resin and then loaded onto a 1 mL HiTrap Heparin column that had been previously equilibrated with Buffer $A_{100}$ after which the column was washed with Buffer $A_{100}$ and the bound proteins were eluted with a 20 mL gradient from Buffer $A_{100}$ to Buffer A containing 1 M NaCl (Buffer $A_{1000}$). The protein-containing fractions were loaded onto a 110 mL (1.6 × 55.5 cm) Superdex 75 column that had been previously equilibrated with Buffer $A_{100}$ and then the column was washed with Buffer $A_{100}$. The Rad27-containing fractions from the Superdex 75 column were subsequently loaded onto a 1 mL MonoS column, washed with Buffer $A_{100}$, and the bound proteins were eluted with a 30 mL linear gradient from Buffer $A_{100}$ to Buffer $A_{1000}$. The resulting purified Rad27 was pooled and stored in small aliquots at −80 °C.

**In vitro MMR assays.** The in vitro MMR repair assay was performed essentially as described[17,18]. One hundred nanograms of circular DNA substrate containing a CC mispair disrupting a PstI site and a nick located 5' of the mispair at the NaeI site was used as a substrate, and the final reaction contained 33 mM Tris pH 7.6,

75 mM KCl, 2.5 mM ATP, 1.66 mM glutathione, 8.3 mM MgCl$_2$, 80 µg/mL BSA, and 200 µM of each dNTP, 290 fmole PCNA, 220 fmole RFC-Δ1N, 390 fmole Msh2-Msh6 or mutant Msh2-Msh6, as indicated, 20 fmole DNA polymerase δ, 1.05 fmole Exo1, as indicated, and 1800 fmole RPA in a 10 µL volume. As indicated in individual experiments, the reactions also contained different amounts of Rad27 that were added at the same time as the other proteins were added, although most reactions contained 5 pmole of Rad27. Where necessary, proteins were diluted in a buffer consisting of 7.5 mM HEPES pH 7.5, 200 mM KCl, 1 mM DTT, 10% (v/v) glycerol, and 0.5 mg/mL BSA prior to addition to the reactions. The reactions were then stopped by the addition of EDTA, Proteinase K (Sigma Aldrich), and glycogen (Thermo Scientific) to final concentrations of 21 mM, 24 µg/mL, and 13.4 µg/mL, respectively, followed by incubation for 30 min at 55 °C. Reactions were extracted with phenol, and the DNA substrate was precipitated with ethanol, followed by digestion with PstI and ScaI. Digested DNA substrate was subjected to electrophoresis on a 0.8% agarose gel run in a buffer containing 40 mM Tris, 20 mM acetic acid, 1 mM EDTA at pH 8.3 (TAE; Biorad) with 0.5 µg/mL ethidium bromide for 45 min at 100 V. Quantitation of the relative amounts of the different DNA species in each individual lane was performed with Alpha Imager HP software (ProteinSimple). Except as indicated in the legend to Supplementary Fig. 2b, all experiments were performed a minimum of 3 times and representative examples of each experiment are presented. When necessary, bar graphs were constructed in GraphPad Prism software.

**Ligase/supercoiling assays**. One hundred nanograms of circular DNA substrate containing a CC mispair disrupting a PstI site and a nick located 5' of the mispair at the NaeI site[17,18] or a dephosphorylated version were used as substrate. Dephosphorylation was performed by incubating the substrate with rSAP (New England Biolabs) at 37 °C for 1 h, followed by incubation for 5 min at 65 °C, phenol extraction, and ethanol precipitation. In vitro MMR assays were then performed as described above. Two different amounts of Rad27, 5 and 2.5 poles, were used. After the DNAs were purified from the repair reactions, the DNA from each repair reaction was split into two equal parts and treated as follows: (1) The repair products were digested with PstI and ScaI for 1 h at 37 °C and then analyzed by electrophoresis on a 0.8% agarose gel run in TAE containing 0.5 µg/mL ethidium bromide for 45 min at 100 V[17,18]. (2) The repair products were incubated with T4 DNA ligase (New England Biolabs) for 2.5 h at room temperature and then incubated with *E. coli* DNA gyrase (Sigma Aldrich) for 1 h at 37 °C. The DNAs were then analyzed by electrophoresis on a 1.0% agarose gel run in TAE for 45 min at 100 V, which was stained with 0.5 µg/mL ethidium bromide for 30 min and destained in water for 30 min. Quantitation was performed with Alpha Imager HP software (ProteinSimple).

**Microscopy**. Cells were grown in synthetic complete medium to log phase and examined by live imaging essentially as previously described[30,49] except using an ECHO Revolve epifluorescence microscope with an Olympus PlanApo N 60 × /1.42 Oil Ph3 immersion objective. GFP fluorescence was detected using a Revolve GFP Cube and ECHO Pro software. Images were saved as TIFF files and visualized using Adobe Photoshop to manually score cells for the presence of GFP foci. Figures were prepared in Adobe Photoshop, keeping processing parameters constant within each experiment. The reported experiments were done twice and a minimum of 8 independent fields were captured and quantified per strain analyzed.

**Reporting summary**. Further information on research design is available in the Nature Research Reporting Summary linked to this article.

## Data availability

The data generated by this study are provided in the main text and figures and the accompanying Supplementary Information and are available from the corresponding author upon reasonable request. Source data are provided with this paper.

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

## Acknowledgements
This study was supported by NIH grant R01 GM50006. We would like to thank Matthew DuPrie for measuring the mutation rates of the *msh2Δ* and *pms1Δ* single mutants.

## Author contributions
R.D.K. conceived the overall experimental design and F.A.C., C.D.P. and R.D.K. participated in the design of biochemical and genetic experiments. C.D.P. designed the *exo1Δ440-702* mutation and analyzed all of the genetic data. K.A.T. and K.N. constructed the strains used for mutation rate assays and performed all of the mutation rate assays. B.-Z.L. constructed all of the strains used for Pms1 foci analysis and performed all of the Pms1 foci experiments. F.A.C. purified independent preparations of the MMR proteins and performed all of the biochemical MMR assays. N.B. constructed the Rad27 over-production plasmid and purified Rad27 and most of the proteins used in the MMR assays. F.A.C., C.D.P., and R.D.K. wrote the paper and all of the authors revised and modified the paper.

## Competing interests
The authors declare no competing interests.
