## [Peer Review File · Nature Communications]

Rad27 and Exo1 function in different excision pathways for Mismatch Repair in *Saccharomyces cerevisiae*Reviewers' Comments:

Reviewer #1:

Remarks to the Author:

Calil et al. address an important question in the DNA mismatch repair field-why do cells lacking Exo1, the primary (if not only identified so far) 5' to 3' exonuclease that acts in mismatch repair, have such mild mutator phenotypes? In addition to previous work providing evidence for Exo1-independent mismatch repair pathways the authors hypothesized, based on the similar biochemical activities of Exo1 and Rad27/Fen1 nuclease that Rad27 could play a minor/redundant role in mismatch repair. Previous support for this idea was lacking because yeast strains lacking Rad27 display a variety of chromosome instability phenotypes including large insertion/deletion mutations and insertions in dinucleotide repeats that are inconsistent with a mismatch repair role. In addition, rad27 null mutants display a weak mutator phenotype in a frameshift assay (hom3-10) that detects single nucleotide frameshifts and provides a sensitive readout for mutants defective in mismatch repair. Epistasis work was not performed with exo1 and rad27 null mutations because the double mutant is lethal.

To explore roles for Rad27 in mismatch repair, the authors used the hom3-10 frameshift assay and took advantage of an exo1(delta440-702) mutation that they showed previously disrupted Exo1-Msh2 interactions but did not confer lethality in the rad27null background. They also used a pms1-A99V mutation that they had shown previously inactivated Exo1-independent mismatch repair.

The key observations were:

The authors observed synergistic increases in mutation rate in the double exo1(delta440-702) rad27delta mutant compared to the single mutants. This synergistic increase was seen in both PMS1 and pms1-A99V strains.

exo1(delta440-702) and rad27null showed similarly elevated levels of Pms1 foci that were not dramatically increased in the double mutant background. Importantly, foci formation in all of these strains was dependent on Msh2. This is a nice observation, though for this reader it's not clear if the presence of foci reflects a role for Rad27 in mismatch repair (one might have expected a significantly higher level of foci in the exo1(delta440-702) rad27null double compared to the single mutants) or a situation where mutations in exo1 and rad27 confer a wide range of DNA lesions, some of which are recognized by the mismatch repair machinery.

Lastly the authors reconstituted a 5' directed mismatch repair pathway involving Rad27 using methodologies they had developed previously. In this work they observed strand displacement activities of DNA polymerase delta in the absence of Msh2-Msh6 and Exo1 that appear to yield repair product. This product was no longer seen when Msh2-Msh6 was added but was reconstituted when Rad27 or Exo1 were added. For Rad27 addition, it appeared that Rad27 acted to remove the 5' flaps created by strand displacement by DNA polymerase delta, suggesting that Rad27 played a role in activating DNA polymerase delta when Msh2-Msh6 was present.

Comments:

A very nice set of observations are presented with clear mechanistic interpretations that open a new window into alternative mismatch repair pathways. I have a few suggested experiments below.

1. The experiments involving the Msh2-Msh6, Msh2-msh6-F337A and Msh2-msh6-FFAA are really interesting and left me with the following questions: Couldn't inhibition of polymerase delta strand displacement activity by Msh2-Msh6 be an issue of titrating away PCNA? The inhibition by Msh2-Msh6 is lost when including a Msh2-Msh6 complex defective in interactions with PCNA (Msh2-msh6-FFAA) but is not lost when a complex defective in mismatch recognition but proficient in PCNA interactions (Msh2-msh6-F337A) is included (Figure 6). It seems important to perform titrations of PCNA levels in

the Polymerase delta only reaction, as well as adding increasing amounts of PCNA in a reaction containing Polymerase delta and Msh2-Msh6. In addition, it could be useful (but not critical) to perform studies with rad27 protein defective in its interaction with PCNA.

2. The Bambara lab has done some nice studies linking Rad27 flap cleavage to ligation by Cdc9. Did the authors test if the repair products could be converted to closed circular at different times in the Rad27 addition reactions? Such observations would strengthen their model for Rad27 acting to remove the 5' flaps created by Polymerase delta strand displacement.

3. The authors cited previous work in which a synergistic increase in mutation rate was observed in rad27 null mismatch repair null double mutants. With that said it seems worth testing the exo1(delta440-702) rad27delta, pms1delta to see if epistasis (consistent with acting in MMR pathways) is observed in the hom3-10 assay.

Minor issues

1. It would help the reader follow the biochemical studies better if cartoons were included in Figure 4 depicting the strand displacement steps catalyzed with Polymerase delta alone. This would allow the reader to better interpret the model in Figure 7.

2. Page 10: Lines 13 and 14. There seem to be some words missing.

Reviewer #2:

Remarks to the Author:

To date, Exo1 is the only resection activity that has been unequivocally shown to function during MMR in yeast, and yet EXO1 deletion causes only a weak mutator phenotype. Early and controversial MMR studies implicated the Rad27 flap endonuclease in MMR and subsequent studies have shown that Exo1 and Rad27 have similar biochemical activities in terms of promoting nick translation. Because loss of both proteins is lethal in yeast, however, a direct test of their redundancy during MMR has not been previously possible. The Kolodner lab identified an exo1 allele (exo1 Δ 440-702) that eliminates its MMR function but not its essential role in a rad27 Δ background, which has allowed the redundancy of these proteins to finally be examined. The genetic experiments support the existence of three separate pathways for mismatch excision and very importantly, the Rad27 pathway was reconstituted in vitro. Altogether, the authors convincingly demonstrate a role of Rad27 in yeast MMR and their data support the existence of three separate pathways for mismatch excision: Exo1 excision past the mismatch, strand displacement past the mismatch that is coupled to Rad27 removal of the resulting flap, and repetitive nicking by Pms1-Mlh1. This is a very nice story and only a few minor comments are given below.

Figure 1 could be moved to the supplement.

Please describe what RFC Δ 1N is.

p. 4, line 17 – repair “of” a mispair

p. 10, line 13 – something is missing from the sentence that begins here

p. 12, line 15 “and” hence

Reviewer #3:

Remarks to the Author:

The describes studies that conclude the Rad27/Fen1 flap-endonuclease is a new redundant pathway of mismatch repair (MMR) strand-specific excision. The experiments are a combination of genetic

mutator studies and biochemical analysis with similar proteins. Both are rather underwhelming and could be significantly improved.

For example, the genetic analysis examined the synergistic effects of the pms1(A99V) mutation. This mutation in the ATPase region has ~20% of the endonuclease activity and ~80% of the MMR activity in vitro (ref. 21). Thus, one could question whether this is the right mutation to examine redundancy with the ExoI-independent pathway. The Kolodner group has previously shown that activation of the Pms1 endonuclease is essential for the ExoI-independent pathway (ref. 1). In that paper the authors have identified a number of mutations in PCNA [(Pol30; e.g. pol(K217E)] and Pms1 [e.g. pms(E707A)] that are incapable of activating the endonuclease and are clear genetic indicators of the contribution(s) of the ExoI-independent pathway. Combined with exoI Δ 440-702 and the Δ rad27, one of these ExoI-independent pathway mutations would be a much better gauge of redundancy in MMR pathways.

The biochemistry also needs some fortification. The involvement of Rad27 protein is best exemplified by comparing Fig. 3 and 4. In a reaction that has all the ExoI-dependent components (RPA, PCNA, RFC, and dNTPs) the addition of Pold, Msh2-Msh6 and ExoI results in about 17% repair products (Fig. 3B). Substituting Rad27 for ExoI starts as a very small high smear that looks identical when only Pold and Msh2-Msh6 are included. With more Rad27, this resolves into what appear to be a large amount (81%) of bona fide repair products (last lane in Fig. 3B). It is claimed by the authors that this is a displaced DNA strand initiated by Pold and trimmed by Rad27 to a repair product. However, the total DNA looks to be significantly more than the input comparing Lane 2 to that last lane? And the authors seem to combine these displaced strand products with the repair product in their quantifications? Figure 4 seems to show that Msh2-Msh6 inhibits these strand displacement products, which even in the absence of Msh2-Msh6 may be resolved to repair products? Fig. 5 also shows significant "repair" in the absence of Msh2-Msh6 (the 1.8 kb product only; eliminating what the authors call strand displacement). The authors might want to parse out the "strand displacement" from the "1.8 kb product" for some illumination here. The argument that binding of Msh2-Msh6 to PCNA is required for appropriate resolution might be correct. But one could also imagine some type of interference by the so-called sliding clamps in the absence of a complete reaction.

The absence of Mlh1-Pms1 in these reactions is glaring! One of the models has suggested the formation of a mismatch-dependent complex at the mismatch. Combined with the genetic problems discussed above, the inclusion of Mlh1-Pms1 would seem essential. This does not mean a complete study of Mlh1-Pms2 nicking in any potential Rad27-dependent reaction. However, the authors have previous published work with a protein equivalent to the pms(E707A) mutation that they discovered, Mlh1-Pms1(E707K). It seems like this should at the very least be tested for its effect on the complete reaction?

The models seem a bit random in their consideration of Mlh1-Pms1 in the various pathways. There is no involvement of Mlh1-Pms1 in the Rad27 pathway, while there is no involvement of Msh2-Msh6 in the ExoI-dependent pathway. Neither of these is likely to be true. For sure the ExoI-dependent pathway does the entire reaction without Mlh1-Pms2, but absolutely requires Msh2-Msh6. The Kolodner group showed this in yeast about 10 years after it was shown in humans.

Lastly, the manuscript could use some editing of run-on sentences and consistent use of terminology (e.g. Rad27 or Rad27/Fen1).

Point-by-point response to the reviewers' comments

General

We would like to thank the reviewers for all of their helpful comments. We have tried to address all of the comments by modifying our manuscript and including new data requested by the reviewers. In addition, we have done some general editing of the manuscript. Below we list each reviewer comment followed by a description of the changes and additions we have made to address each comment; the descriptions of our modifications are in red text and the relevant sections of the manuscript are highlighted in yellow.

Reviewer #1 (Remarks to the Author):

Calil et al. address an important question in the DNA mismatch repair field-why do cells lacking Exo1, the primary (if not only identified so far) 5' to 3' exonuclease that acts in mismatch repair, have such mild mutator phenotypes? In addition to previous work providing evidence for Exo1-independent mismatch repair pathways the authors hypothesized, based on the similar biochemical activities of Exo1 and Rad27/Fen1 nuclease that Rad27 could play a minor/redundant role in mismatch repair. Previous support for this idea was lacking because yeast strains lacking Rad27 display a variety of chromosome instability phenotypes including large insertion/deletion mutations and insertions in dinucleotide repeats that are inconsistent with a mismatch repair role. In addition, rad27 null mutants display a weak mutator phenotype in a frameshift assay (hom3-10) that detects single nucleotide frameshifts and provides a sensitive readout for mutants defective in mismatch repair. Epistasis work was not performed with exo1 and rad27 null mutations because the double mutant is lethal.

To explore roles for Rad27 in mismatch repair, the authors used the hom3-10 frameshift assay and took advantage of an exo1(delta440-702) mutation that they showed previously disrupted Exo1-Msh2 interactions but did not confer lethality in the rad27null background. They also used a pms1-A99V mutation that they had shown previously inactivated Exo1-independent mismatch repair.

The key observations were:

*The authors observed synergistic increases in mutation rate in the double *exo1(delta440-702) rad27delta* mutant compared to the single mutants. This synergistic increase was seen in both *PMS1* and *pms1-A99V* strains.*

*exo1(delta440-702) and rad27null showed similarly elevated levels of Pms1 foci that were not dramatically increased in the double mutant background. Importantly, foci formation in all of these strains was dependent on Msh2. This is a nice observation, though for this reader it's not clear if the presence of foci reflects a role for Rad27 in mismatch repair (one might have expected a significantly higher level of foci in the *exo1(delta440-702) rad27null* double compared to the single mutants) or a situation where mutations in *exo1* and *rad27* confer a wide range of DNA lesions, some of which are recognized by the mismatch repair machinery.*

Lastly the authors reconstituted a 5' directed mismatch repair pathway involving Rad27 using methodologies they had developed previously. In this work they observed strand displacement activities of DNA polymerase delta in the absence of Msh2-Msh6 and Exo1 that appear to yield repair product. This product was no longer seen when Msh2-Msh6 was added but was reconstituted when Rad27 or Exo1 were added. For Rad27 addition, it appeared that Rad27 acted to remove the 5' flaps created by strand displacement by DNA polymerase delta, suggesting that Rad27 played a role in activating DNA polymerase delta when Msh2-Msh6 was present.

Comments:

A very nice set of observations are presented with clear mechanistic interpretations that open a new window into alternative mismatch repair pathways. I have a few suggested experiments below.

- 1. The experiments involving the *Msh2-Msh6*, *Msh2-msh6-F337A* and *Msh2-msh6-FFAA* are really interesting and left me with the following questions: Couldn't inhibition of polymerase delta strand displacement activity by *Msh2-Msh6* be an issue of titrating away PCNA? The inhibition by *Msh2-Msh6* is lost when including a *Msh2-Msh6* complex defective in interactions with PCNA (*Msh2-msh6-FFAA*) but is not lost when a complex defective in mismatch recognition but proficient in PCNA interactions (*Msh2-msh6-F337A*) is included (Figure 6). It seems important to perform titrations of PCNA levels in the Polymerase delta only reaction, as well as adding*

increasing amounts of PCNA in a reaction containing Polymerase delta and Msh2-Msh6. In addition, it could be useful (but not critical) to perform studies with rad27 protein defective in its interaction with PCNA.

Based on our original data showing that the Msh2-Msh6-FFAA was unable to reduce strand displacement synthesis by DNA polymerase delta, we suspected that PCNA titration might be the underlying mechanism. To confirm this, we have performed titrations that have reduced the levels of Msh2-Msh6 (**see new Supplementary Figure 2A**) and increased the levels of PCNA (**see new Supplementary Figure 2B**) in reactions containing DNA polymerase delta, RFC, PCNA, and Msh2-Msh6. Consistent with our (and the reviewer's) hypothesis of sequestration of PCNA by Msh2-Msh6, either decreasing the levels of Msh2-Msh6 or increasing the levels of PCNA was capable of restoring strand displacement synthesis. These new results are described in the manuscript (**see the new 2nd paragraph page 11**).

2. The Bambara lab has done some nice studies linking Rad27 flap cleavage to ligation by Cdc9. Did the authors test if the repair products could be converted to closed circular at different times in the Rad27 addition reactions? Such observations would strengthen their model for Rad27 acting to remove the 5' flaps created by Polymerase delta strand displacement.

Although introduction of the Cdc9 DNA ligase into the repair reactions is beyond the scope of this manuscript and because we do not currently have any Cdc9 protein, we have sought to investigate the reviewer's question of whether repair products can be ligated into closed circular DNAs using T4 DNA ligase (**see the new Supplementary Figure 3 and the new 3rd paragraph on page 11 and the new 2nd paragraph on page 12**). To test if the reaction products were ligatable, we developed a protocol in which products were deproteinated and then treated with a combination of T4 DNA ligase and E. coli DNA gyrase. In these reactions, ligatable nicked circles are covalently closed by ligase and then converted to supercoiled circles by gyrase (**see the new Supplementary Figure 3**). Importantly, in order to test the ability of the products to be ligated, it was necessary to ensure that the starting mispair-containing substrate could not be ligated. To do this, we developed a procedure to dephosphorylate the nicked substrate in order to remove the 5' phosphate required for the ligation reaction. This dephosphorylated product was found to be proficient for mispair-directed repair as dephosphorylation did not affect the 3' hydroxyl at the nick required as a primer terminus by

DNA polymerase delta. Consistent with the reviewer's hypothesis, a large proportion of the flap-cleaved products could be sealed by DNA ligase.

3. The authors cited previous work in which a synergistic increase in mutation rate was observed in *rad27* null mismatch repair null double mutants. With that said it seems worth testing the *exo1(delta440-702) rad27delta, pms1delta* to see if epistasis (consistent with acting in MMR pathways) is observed in the *hom3-10* assay.

Previously it has been demonstrated that a deletion of *RAD27* caused a weak mutator phenotype and was epistatic to a deletion of *MSH2* in the *hom3-10* frameshift reversion assay (PMID: 9008166), suggesting that Rad27 could play a minor role in MMR. We have now constructed the *msh2Δ exo1Δ440-702 rad27Δ* and *pms1Δ exo1Δ440-702 rad27Δ* triple mutants and measured their mutation rates in both the *hom3-10* and *lys2-10A* frameshift reversion assays (see additions to Table 1 and the first paragraph on page 7). We found that the mutation rates of the triple mutants were slightly higher (1.5-4 fold higher) than strains possessing only a deletion of *MSH2* or *PMS1*, although some of the differences between the mutation rates were not significant based on overlapping 95% confidence intervals. These data do not provide strong evidence for a role for Rad27 in a non-MMR pathway that suppresses frameshift mutations in the *hom3-10* assay; however, the small increase in mutation rate could be consistent with a role of *RAD27* in reducing the number of mispairs to be corrected by removal of the portion of Okazaki fragments synthesized by DNA polymerase alpha, which lacks a proofreading activity, as has been shown in mammalian systems (PMID: 25921062).

Minor issues

1. It would help the reader follow the biochemical studies better if cartoons were included in Figure 4 depicting the strand displacement steps catalyzed with Polymerase delta alone. This would allow the reader to better interpret the model in Figure 7.

A new scheme was added to the current **Figure 2** showing the strand displacement catalyzed by DNA polymerase delta and formation of a 5' flap, which is cleaved by Rad27, resulting in a PstI/Scal digestible and nick translated product.

2. Page 10: Lines 13 and 14. There seem to be some words missing.

Corrected.

Reviewer #2 (Remarks to the Author):

To date, *Exo1* is the only resection activity that has been unequivocally shown to function during MMR in yeast, and yet *EXO1* deletion causes only a weak mutator phenotype. Early and controversial) MMR studies implicated the *Rad27* flap endonuclease in MMR and subsequent studies have shown that *Exo1* and *Rad27* have similar biochemical activities in terms of promoting nick translation. Because loss of both proteins is lethal in yeast, however, a direct test of their redundancy during MMR has not been previously possible. The Kolodner lab identified an *exo1* allele (*exo1* Δ 440-702) that eliminates its MMR function but not its essential role in a *rad27* Δ background, which has allowed the redundancy of these proteins to finally be examined. The genetic experiments support the existence of three separate pathways for mismatch excision and very importantly, the *Rad27* pathway was reconstituted in vitro. Altogether, the authors convincingly demonstrate a role of *Rad27* in yeast MMR and their data support the existence of three separate pathways for mismatch excision: *Exo1* excision past the mismatch, strand displacement past the mismatch that is coupled to *Rad27* removal of the resulting flap, and repetitive nicking by *Pms1-Mlh1*. This is a very nice story and only a few minor comments are given below.

1. Figure 1 could be moved to the supplement.

As requested, we have moved Figure 1 to the supplement (see new Supplementary Figure 1).

2. Please describe what *RFC* Δ 1N is.

RFC- Δ 1N is a version of Replication Factor C in which the ligase homology domain of *Rfc1* was deleted to allow for overexpression in *E. coli*. We have now described this in the results section of the manuscript (see 3rd paragraph, page 8).

p. 4, line 17 – repair “of” a mispair

Corrected.

p. 10, line 13 – something is missing from the sentence that begins here

Corrected.

p. 12, line 15 “and” hence

Corrected.

Reviewer #3 (Remarks to the Author):

The describes studies that conclude the Rad27/Fen1 flap-endonuclease is a new redundant pathway of mismatch repair (MMR) strand-specific excision. The experiments are a combination of genetic mutator studies and biochemical analysis with similar proteins. Both are rather underwhelming and could be significantly improved.

For example, the genetic analysis examined the synergistic effects of the pms1(A99V) mutation. This mutation in the ATPase region has ~20% of the endonuclease activity and ~80% of the MMR activity in vitro (ref. 21). Thus, one could question whether this is the right mutation to examine redundancy with the Exo1-independent pathway. The Kolodner group has previously shown that activation of the Pms1 endonuclease is essential for the Exo1-independent pathway (ref. 1). In that paper the authors have identified a number of mutations in PCNA [(Pol30; e.g. pol(K217E)] and Pms1 [e.g. pms(E707A)] that are incapable of activating the endonuclease and are clear genetic indicators of the contribution(s) of the Exo1-independent pathway. Combined with exo1Δ440-702 and the Δrad27, one of these Exo1-independent pathway mutations would be a much better gauge of redundancy in MMR pathways.

A hallmark of the previously identified mutations that appear to cause defects in Exo1-independent MMR is that they all caused reduced, but not eliminated, Mlh1-Pms1 endonuclease activation (see for instance PMIDs 11438669, 24204293, 24981171). Thus, the same reduced Mlh1-Pms1 nuclease activity and partial MMR defect caused by the pms1-A99V mutation is also

observed with mutations in *POL30*, such as the *pol30-K217E* mutation that the reviewer is suggesting we study (PMID 24981171). Unlike *pms1-A99V*, mutations in *POL30*, which encodes PCNA, also potentially cause other defects that could complicate their use in analysis of Rad27-dependent MMR; Rad27 is recruited to DNA by PCNA and a number of the Exo1-independent MMR mutations in *POL30* cause defects in PCNA trimerization and DNA retention that could affect the role of Rad27 in MMR independent of any effects in activating the Mlh1-Pms1 endonuclease.

In spite of these complications, we constructed the *rad27Δ pol30-K217E* double mutation and tested it in frameshift reversion assays (**see 2nd paragraph page 7**). Consistent with the genetic interactions causing increased mutation rates when combining *rad27Δ* and *pms1-A99V*, *exo1Δ* and *pms1-A99V*, and *exo1Δ* and *pol30-K217E*, the *pol30-K217E* mutation showed increased mutation rates when combined with the *rad27Δ* mutation. Thus, the genetic interactions seen with both the *pms1-A99V* and *pol30-K217E* mutations support the model for three excision pathways in MMR shown in Fig. 6 (previously numbered Fig. 7).

The biochemistry also needs some fortification. The involvement of Rad27 protein is best exemplified by comparing Fig. 3 and 4. In a reaction that has all the Exo1-dependent components (RPA, PCNA, RFC, and dNTPs) the addition of Pold, Msh2-Msh6 and Exo1 results in about 17% repair products (Fig. 3B). Substituting Rad27 for Exo1 starts as a very small high smear that looks identical when only Pold and Msh2-Msh6 are included. With more Rad27, this resolves into what appear to be a large amount (81%) of bona fide repair products (last lane in Fig. 3B). It is claimed by the authors that this is a displaced DNA strand initiated by Pold and trimmed by Rad27 to a repair product. However, the total DNA looks to be significantly more than the input comparing Lane 2 to that last lane? And the authors seem to combine these displaced strand products with the repair product in their quantifications?

Although the total amount of DNA might vary somewhat between lanes, small errors in loading or increased amounts of DNA due to DNA synthesis do not affect the results because the percentages of repaired products is quantified on a lane-by-lane basis analyzing only the DNA present in each individual lane. We have clarified this in the manuscript (**see 2nd paragraph page 19**).

Further, we quantified the percentage of repaired product as including all of those products that can be cleaved by PstI, which indicates repair of the mispair. These products include both the strand-displacement bands and the mature repair products, in which the 5' flap has been cleaved. This is also more accurate, as in some lanes it is not possible to clearly distinguish between the strand-displacement repair products and the mature repair products. This has been clarified in the text (**see the legend for Figure 2-5 and Supplementary Figures 2 and 3**).

Figure 4 seems to show that Msh2-Msh6 inhibits these strand displacement products, which even in the absence of Msh2-Msh6 may be resolved to repair products? Fig. 5 also shows significant "repair" in the absence of Msh2-Msh6 (the 1.8 kb product only; eliminating what the authors call strand displacement). The authors might want to parse out the "strand displacement" from the "1.8 kb product" for some illumination here. The argument that binding of Msh2-Msh6 to PCNA is required for appropriate resolution might be correct. But one could also imagine some type of interference by the so-called sliding clamps in the absence of a complete reaction.

In the absence of Msh2-Msh6, only partial products are resolved to mature repair products (product of Rad27 cleavage), which shows that the presence of Msh2-Msh6 promotes the formation of fully mature repair products. In addition to the effects of the Msh2-Msh6-FFAA mutant complex, we have now further verified the role of the Msh2-Msh6 interaction with PCNA by performing titrations that have reduced the levels of Msh2-Msh6 (**see new Supplementary Figure 2A**) and increased the levels of PCNA (**see new Supplementary Figure 2B**).

The absence of Mlh1-Pms1 in these reactions is glaring! One of the models has suggested the formation of a mismatch-dependent complex at the mismatch. Combined with the genetic problems discussed above, the inclusion of Mlh1-Pms1 would seem essential. This does not mean a complete study of Mlh1-Pms2 nicking in any potential Rad27-dependent reaction. However, the authors have previous published work with a protein equivalent to the pms(E707A) mutation that they discovered, Mlh1-Pms1(E707K). It seems like this should at the very least be tested for its effect on the complete reaction?

Based on the updated model figure (**see the updated Fig. 6**) and published work on the Mlh1-Pms1/Pms2 endonuclease, we would suggest that the major role of Mlh1-Pms1 in Rad27-dependent MMR reactions will likely be very similar to its observed role in Exo1-dependent

MMR reactions. In these reactions Mlh1-Pms1 primarily generates 5' nicks on substrates with pre-existing 3' nicks. Other models for Mlh1-Pms1 action in MMR, including more complex ones, have been proposed as well. Our study of Rad27 in MMR is already an extensive study that clearly established a role for Rad27 in a mismatch excision pathway in MMR. Furthermore, in our hands, there are mechanistic complexities in the role of Mlh1-Pms1 in MMR that are not yet well understood. For these reasons, we do not think it is realistic to in addition reconstitute an Mlh1-Pms1 dependent, Rad27 dependent reaction and study its mechanism here, as this is not as simple as the reviewer implies. Thus, in our view, the analysis of Mlh1-Pms1 in the Rad27-dependent MMR reactions constitutes an independent study that is outside the scope of the current manuscript.

The models seem a bit random in their consideration of Mlh1-Pms1 in the various pathways. There is no involvement of Mlh1-Pms1 in the Rad27 pathway, while there is no involvement of Msh2-Msh6 in the Exo1-dependent pathway. Neither of these is likely to be true. For sure the Exo1-dependent pathway does the entire reaction without Mlh1-Pms2, but absolutely requires Msh2-Msh6. The Kolodner group showed this in yeast about 10 years after it was shown in humans.

We have updated the Fig 6 (previously numbered Fig 7) to reflect potentially the recruitment of fewer Mlh1-Pms1 molecules in the Exo1- and Rad27-dependent pathways. It is worthy of note that increased numbers of Mlh1-Pms1 foci are observed in *exo1* mutants suggesting that at least some of the Exo1-independent pathways are associated with the recruitment of multiple Mlh1-Pms1 molecules (see PMID 22118461). For *in vitro* substrates with pre-existing 5' nicks (but not 3' nicks), Mlh1-Pms1 is not required for repair. In contrast, all of the genetics data indicates that both Mlh1-Pms1 and its endonuclease activity are absolutely required for all MMR. This discrepancy has not been fully resolved, though we have recently proposed that the *in vivo* requirement may be due to competition between MMR using nicks to direct excision-repair and DNA ligation in removing replication bound nicks (see PMID: 33417883, 34171627). Thus, we have further modified model figure and figure legend (**see the new Fig 6**) to emphasize that Exo1 and DNA polymerase delta could be initiating excision or strand displacement, respectively, from a pre-existing nick or one generated by Mlh1-Pms1.

Lastly, the manuscript could use some editing of run-on sentences and consistent use of terminology (e.g. Rad27 or Rad27/Fen1).

We have carefully gone through the manuscript and corrected these issues.

Reviewers' Comments:

Reviewer #1:

Remarks to the Author:

The authors have fully addressed my queries through additional experimentation (POL30/Msh2-Msh6 titration, testing whether the Rad27-dependent nicks can be ligated, epistasis analysis involving *exo1delta440-702*, *rad27delta*, *pms1/msh2 delta*). It's a really nice study.

Reviewer #3:

Remarks to the Author:

The authors have adequately addressed my concerns.

Response to Reviewer's Comments

We would like to thank the reviewers for their efforts and their helpful comments. We are pleased that the reviewers feel we have addressed all of their concerns and that they are supportive of our study.

Reviewer #1 (Remarks to the Author):

The authors have fully addressed my queries through additional experimentation (POL30/Msh2-Msh6 titration, testing whether the Rad27-dependent nicks can be ligated, epistasis analysis involving *exo1delta440-702*, *rad27delta*, *pms1/msh2 delta*). It's a really nice study.

No changes were requested.

Reviewer #3 (Remarks to the Author):

The authors have adequately addressed my concerns.

No changes were requested.